# A Publicly Verifiable E-Voting System Based on Biometrics

Jinhui Liu [1,2], Tianyi Han [1], Maolin Tan [1], Bo Tang [1,2,*,†], Wei Hu [1,†] and Yong Yu [1]

[1]  School of Cyber Security, Northwest Polytechnical University, Xi'an 710072, China; jh.liu@nwpu.edu.cn (J.L.); hantianyi127@mail.nwpu.edu.cn (T.H.); tanmaolin@mail.nwpu.edu.cn (M.T.); weihu@nwpu.edu.cn (W.H.); yuyong@snnu.edu.cn (Y.Y.)

[2]  Research & Development Institute, Northwest Polytechnical University, Shenzhen 518057, China

[*]  Correspondence: tangbo@nwpu.edu.cn

[†]  These authors contributed equally to this work.

**Abstract:** Voters use traditional paper ballots, a method limited by the factors of time and space, to ensure their voting rights are exercised; this method requires a lot of manpower and resources. Duplicate voting problems may also occur, meaning the transparency and reliability of the voting results cannot be guaranteed. With the rapid developments in science and technology, E-voting system technology is being adopted more frequently in election activities. However, E-voting systems still cannot address the verifiability of the election process; the results of a given election and the credibility of the host organization will be questioned if the election's verifiability cannot be ensured. Elections may also pose a series of problems related to privacy, security, and so on. To address these issues, this paper presents a public, and verifiable E-voting system with hidden statistics; this system is based on commitment, zk-SNARKS, and machine learning. The system can deal with a large number of candidates, complex voting methods, and result functions in counting both hidden and public votes and can satisfy the requirements of verifiability, privacy, security, and intelligence. Our security analysis shows that our scheme achieves privacy, hidden vote counting and verifiability. Our performance evaluation demonstrates that our system has reasonable applications in real scenarios.

**Keywords:** commitment; zero-knowledge proof; E-voting system; convolutional neural network; face recognition





## 1. Introduction

Election is a procedure with recognized rules that involves the act of electing, by means of voting, one or more persons to hold a certain office or to represent a certain group [1,2]. Elections are not only common in the political sphere but, with the development of democracy, they are also used in other areas; examples of this are the election of persons to the board of directors of a company, the election of student union leaders, and so on. According to the rules of an election, after a credible organization has announced the information of the eligible candidates, the method of the election, and the rules of vote counting, voters with the franchise are generally required to cast their votes at a specified time and place. After the voting has taken place, the credible organization will count the eligible votes. In the end, the candidate who receives the most votes wins the election.

As one of the most important behaviors in the electoral process, voting methods have progressed in parallel with the development of the electoral system, from initially simple methods (such as standing in line to vote in public, and hand raising) to the implementation of ballot boxes (or similar devices), which serve to protect the privacy of voting citizens. In addition, vote-counting methods have also evolved, with cumulative voting and instant run-off systems [3] being used in different scenarios.

The way in which we safeguard verifiability is an important issue for current electronic voting processes; this issue has arisen due to the discrepancy between practical applications and peoples' goals, needs, and expectations. Verifiability refers to the extent to which an idea, theory, hypothesis or conclusion can be confirmed or falsified. The concept of

verifiability, as it pertains to the electoral process, means that voters should be able to rigorously verify in some way that their eligible votes have been correctly counted after the results have been announced.

To confront this problem, generally speaking, after the announcement of the final winner of the election, the fairness, impartiality, and authenticity of the entire election process can be ensured by publishing a complete tally of all counted votes; at the same time, the verifiability of the votes can be ensured through the publication of the count-related information [4]. However, information on the actual processes of elections is generally not published in this way because this approach may prompt a series of problems, such as the disclosure of voters' personal data; if these problems are induced, the authority of the final winner may be negatively affected. This problem can be solved by concealing votes. Vote hiding refers to hiding some (or all) of the votes when announcing the results of an election so as to protect the privacy of the voters. However, in adopting the method of vote concealment, the verifiability of the voting process cannot be effectively guaranteed. Therefore, the concept of voting concealment alone cannot guarantee verifiability.

Yet another important issue faced during the voting process is ensuring the verifiability of the voters' identities so that the fairness and legitimacy of elections is guaranteed. In some countries, elections are the cornerstone of democratic politics, and the votes of voters determine the leaders of the government and the direction of its policies; therefore, the fairness and legitimacy of elections are very important. Voter identity verification prevents electoral fraud and acts of interference (including multiple voting, false voter registration, and voter identity theft). Failure to verify the identity of voters may lead to the manipulation of election results through unruly acts and fraudulent means, thereby undermining social stability and justice. In addition, voter identity verification can also ensure the legitimacy of voters' franchise. Voter eligibility is usually restricted by laws and policies, such as age, nationality, place of residence, and other conditions. The verification of voter identity can ensure that only those who fulfill the eligibility criteria can participate in the election, and will prevent the participation of illegitimate voters in polls. The verification of voter identity is therefore a necessary measure taken to maintain the fairness and legitimacy of elections. Methods of verifying voter identity include fingerprint recognition, face recognition, ID recognition, etc., which are applied either individually or in combination in different voting scenarios. Among them, face recognition is the most convenient to implement (compared with other identification methods), and is therefore increasingly used in various identification scenarios.

Therefore, the above two problems can be effectively solved through public vote counting and face recognition methods, respectively.

Within this study, Section 2 details the current status of research on electronic voting systems, and Section 3 introduces some preliminaries of the E-voting system and its properties, some hard problem assumptions, and other cryptographic techniques. In Section 4, we provide the system modeling and security requirements of our constructed and publicly verifiable E-voting system. We design a concrete scheme in Section 5. A security analysis and performance evaluation are provided in Sections 6 and 7. Lastly, we present our conclusions in Section 8.

## 2. Related Works

E-voting systems use electronic technology to conduct an election or voting event. E-voting systems can effectively guarantee verifiability, privacy, and other related factors of the election process. At present, the existing and relevant research studies on E-voting systems at home and abroad are as follows.

*Verifiability of E-voting systems:* Trust in voting (i.e., voters' acceptance of the vote and its results) is highly dependent on the system's ability to protect ballot information. Ensuring that the final results of the E-voting system are trusted by the populace is of great importance because elections without trust (that cannot be verified (or validated)) are unacceptable in democratic societies. To solve this problem, Khlaponin et al. [5] provided an E-voting scheme with adequate verifiability, in order to avoid distrust in the accuracy of vote counting,

ensuring voters' privacy through the provision of fully transparent hardware and software. Suharsono et al. [6] proposed an indicator for improving the verifiability of E-voting systems, knowing that verifiability is one of the most important features in E-voting systems and is the most important factor in improving public acceptance of the results of E-voting. Suharsono et al. [7] proposed an end-to-end verifiability metric to measure the degree of end-to-end verifiability (including end-to-end verifiability metrics) before an election, end-to-end verifiability at the time of an election, end-to-end verifiability after an election, and end-to-end verifiability after the counting of votes. They also determined the location of the degree of verifiability, and determined the range of the degree of verifiability. Cortier et al. [4] provided an in-depth study of verifiability in electronic voting systems.

*Privacy in E-voting systems*: Another important characteristic of E-voting systems is privacy. Rodiana et al. [8] proposed a system design based on public key infrastructure and hash functions alongside key factors in the management of E-voting systems. Within this system, privacy and verifiability are organically combined because the voter's identity and ballot information are not disclosed; meanwhile, the results of the individual's choice can be verified at each stage of the E-voting. Ramchen et al. [9] designed an E-voting system based on the concept of partial counting concealment, which prevents the exposure of private information during the election process. Currently, the concept of counting concealment in E-voting systems is commonly divided into those of full vote count concealment, partial vote count concealment, and open vote count concealment [10,11].

*Blockchain-based E-voting systems*: Blockchain, with its qualities of decentralization and open and transparent data, can solve the problems faced by current E-voting systems; it can make the voting process open and transparent and therefore enhance the trustworthiness of voting results. Mookherji et al. [12] proposed several E-voting methods involving blockchain concepts for trust and transparency; they explored various E-voting protocols, and conducted a comparative study of the performance, security features, and limitations. The Ordinos E-voting system [2,10] is the first provably secure and verifiable full-count-concealing E-voting system, and has been widely used within various voting methods and election scenarios.

*Electronic voting system using biometric technology*: Currently, biometric technology is widely used in various products. G. Revathy et al. proposed an electronic voting scheme using deep learning technology for face recognition. The process of casting a vote was accomplished with the use of blockchain technology and a blind signature mechanism. The main objective of the proposed scheme is to explore the positive effects of security and safety on an online voting system [13]. A study by S. Heiberg et al. carefully considered facial recognition as a possible remote voter recognition measure. They discussed facial recognition technology in relation to voting, covering major architectural decisions and analyzing some unresolved issues including dispute resolution and privacy problems [14]. In addition, S. S. Najam et al. proposed, implemented, and analyzed a new electronic voting system based on a hybrid design involving voter identification based on fingerprint and facial recognition methods. The cross validation of voters during the election process provides better accuracy than single-parameter identification methods. The facial recognition system therein used the Viola Jones algorithm along with the rectangular Haar feature selection method to detect and extract features to develop biometric templates and perform feature extraction during voting [15].

*Other applications of E-voting systems*: Although E-voting methods are efficient and solve the shortcomings of traditional voting systems, they also present new challenges for researchers, such as voter authentication, voter privacy protection, and vote verifiability. Currently, some primitive cryptographic methods, such as blind signatures and ring signatures, have been incorporated into E-voting protocols to provide enhanced security and robustness to E-voting systems. Kurbatov et al. [16] introduced a mechanism for ensuring anonymity to decentralized E-voting systems using ring signatures. Unlike standard signature algorithms, which allow the verification of the unique author of a signature, ring signatures allow the real public key for verification to be hidden among the public

keys of other participants in the system. Canard et al. [17] designed an E-voting system for majority judgment voting in order to implement a majority judgment voting method; however, this system took a relatively long time to complete the same computation tasks as present E-voting systems.

To summarize, although the current field of electronic voting systems is developing rapidly, and various technologies are being implemented, an effective electronic voting system solution that can ensure the protection of user privacy while satisfying verifiability is still lacking. At the same time, the structure of some electronic voting systems is too complex, supporting relatively few voting methods.

Based on the concept of public vote counting and hiding, this paper presents a verifiable electronic voting system design with the concealment of public statistics; this is achieved by applying cryptographic commitment, zero-knowledge proof, and other technologies. This system is intended for elections with both public and hidden vote counting, and can handle a large number of candidates and complex voting methods and result functions while meeting electronic voting systems' requirements of verifiability, privacy, and so on. At the same time, the electronic voting system herein involves face recognition technology in order to further improve security and the unforgeability of user identities.

## 3. Preliminaries

In this section, we focus on the various basic elements and techniques covered in this thesis. In this part, the basic elements and related technologies involved in the E-voting system solution designed in this paper are explained.

### 3.1. Generic Electronic Voting System

The traditional paper-based voting method has two problems.

1. Temporal and spatial limitations: traditional paper ballots are limited by time and space, they consume a lot of manpower and material resources, and are too costly and time consuming.
2. Data security: The contents of the paper ballots, i.e., the voting data, are easily tampered with and falsified, and there is the problem of duplicate voting, which means the transparency and reliability of the data may be called into question.

To address the above problems, E-voting systems are proposed; they involve the use of electronic technologies to conduct elections. They consist of hardware devices and software program that voters can use to cast their votes; the said devices and programs are then used to count the votes and report the results through electronic devices as shown in Figure 1.

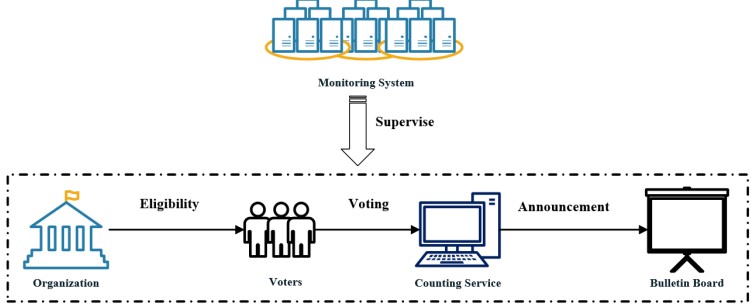

**Figure 1.** General system model of electronic voting.

The use of an electronic voting system in the election process not only allows time and place constraints to be avoided; in addition, through the application of theoretical knowledge of cryptography and other aspects of network security, these systems can also effectively solve problems such as ballot paper security, conduct voting quickly and conveniently, and reduce ballot paper counting errors, while saving a great deal of manpower and financial resources.

However, as an electronic product, E-voting systems have the same problems that other electronic systems face. For example, E-voting systems may be more demanding than traditional paper ballots at the hardware level, and at the software level, system failures may occur due to programming errors or subtle security vulnerabilities.

### 3.2. Hard Problem Assumptions

The discrete logarithm problem is a widely used hard problem assumption in cryptography, which is mainly divided into the discrete logarithm problem over finite fields and the discrete logarithm problem over elliptic curves.

*Discrete logarithm problem over a finite field*: This problem is based on an exponential operation over a finite field, wherein a given element and a power of that element solve for the exponent of that power. Specifically, on a finite domain $GF(p)$ , a given element $g$ and an element $y = g^x$ , where $x$ is unknown, solve for the value of $x$.

*Elliptic Curve Discrete Logarithm Problem* (ECDLP): This problem is based on solving the multiplicity of a given point and a multiplicity of that point, within point group operations on an elliptic curve. Specifically, on an elliptic curve, a given point $P$ and a point $Q = kP$ where $k$ is unknown, solve for the value of $k$.

It is easy to calculate $y$ from $x$ for a discrete logarithm problem that requires, at most, $2 \times \log_2{}^p$ multiplications. However, computing $x$ from $y$ is difficult, and the fastest known algorithm is used for solving a discrete logarithm problem with a time complexity of

$$O(\exp\left((\ln p)^{1/3} \ln(\ln p)\right)^{2/3}).$$

*Bilinear Diffie–Hellman Problem* (BDHP): Given $(P, Q, e(P, Q))$ and $(R, S, e(R, Q), e(P, S))$ in a bilinear mapping, it is hard to solve $e(R, S)$; that is to say, there is no known efficient algorithm that can be used to solve this problem efficiently.

### 3.3. Other Cryptographic Techniques

Although the concept of hidden public counting has solved the problem of privacy protection in current E-voting systems to a certain extent, this concept still fails to protect the verifiability of ballots. This problem can be solved by using homomorphic commitments in cryptographic commitment schemes alongside concise and efficient zero-knowledge proof systems.

#### 3.3.1. Commitment

Commitment is a two-stage interaction protocol involving two parties: a sender and a verifier. Within this scheme, a commitment party sends a committed cipher that will not be changed, and the receiver checks whether the commitment is valid or not.

1.　In the commitment phase, the sender picks a random number for a message m, computes the commitment of the message m, and then sends it to the verifier.
2.　In the open phase, the sender discloses the secret and the random number, and the receiver checks whether the commitment is valid or not.

Using the commitment scheme, in an E-voting system, the voter is the commitment party and the counting agent is the validating party.

1.　Each voter generates a commitment for a specific voting option. This commitment can be publicly verified, but not decrypted, thereby protecting the voter's privacy. After the polls close, the voter can make the commitment public without revealing his or her vote choice.
2.　At the counting stage, the teller only needs to count all the commitments to obtain the number of votes for each option. Since the commitments are hidden, the teller cannot know the specific voting choices of each voter, thus protecting the privacy of the voters.

In applying the commitment scheme, ballot papers within E-voting systems are given the properties of unforgeability, privacy, and integrity during the voting process. At the

same time, voters' ballots can be verified by the correctness and consistency of the ballots, without revealing any private information.

3.3.2. Homomorphic Commitments

Homomorphic commitment is a special commitment scheme with homomorphic function. There exist two plaintexts $m_1$ and $m_2$, the corresponding commitment values are $c_1$ and $c_2$, respectively. Some kind of operation is performed on these two plaintexts to obtain the result $m_3$, and the corresponding commitment $c_3$ can also be calculated using $c_1$ and $c_2$ without decrypting $c_1$ and $c_2$.

Common homomorphic commitment schemes include the Pedersen commitment, Boneh–Boyen signature, and Growth–Sahai commitment. Our construction is based on the Pedersen commitment and Pedersen vector commitment, which are described as follows [18].

The Pedersen commitment can be described as follows.

- *Setup*: Select a group $G$ with order in a large prime number $q$ and with two generate elements $G =< g >=< h >$, and public $(G, h, q)$.
- *Commit*: The committer chooses a random number $r$ as a blind factor (random number), calculates the commitment value = $com_1 = g^v h^r$ mod $q$ for the original message $v$, and then sends $com_1$ to the receiver.
- *Open*: The sender sends $(v, r)$ to the receiver, and after receiving it, the receiver verifies whether $com_1$ is equal to $com_2$ , that is $g^v h^r$ mod $q$, and accepts it if it is equal; otherwise, the promise rejected.

Generally speaking, Pedersen commitments take the following two forms:

- Standard forms over discrete logarithm groups:

$$com = g^v h^r mod q.$$

- Standard forms over elliptic curve groups:

$$com = vG + rH.$$

Homomorphism in the standard form of Pedersen commitment, meaning that if $comm_1$, $comm_2$ are commitments of $v_1$, $v_2$ using two blind factors $r_1, r_2$, respectively, then $comm = comm_1 \times comm_2$ is a commitment to $v_1 + v_2$ using blindness factor $r_1 + r_2$. This means that

$$com = comm(v_1, r_1) \times comm(v_2, r_2) = (g^{v_1} h^{r_1}) \times (g^{v_2} h^{r_2}) = g^{v_1+v_2} h^{r_1+r_2}$$

The Pedersen commitment is based on an additive homomorphism over elliptic curves, meaning that $comm_1$, $comm_2$ are two commitments of $v_1$, $v_2$ using blind factors $r_1$, $r_2$, respectively.

$$comm(v_1, r_1) + comm(v_2, r_2) = v_1 G + r_1 H + v_2 G + r_2 H = comm(v_1 + v_2, r_1 + r_2)$$

The Pedersen vector commitment may be described as follows.

In practical E-voting systems, because the voting process will involve a large amount of data (for example, in the many voting options) if only a Pedersen commitment is used, a separate commitment will need to be made for each piece of data. This will generate a lot of independent Pedersen commitments $com(v_i, r_i)$ . This approach will result in a waste of computational resources and will also affect computational efficiency.

To solve these problems, the Pedersen vector commitment was proposed by Torben Pedersen in 1991 [19]. The core idea of the Pedersen vector commitment is that it can be used for an original information vector $v_1, v_2, \ldots, v_N$; the commitment to this vector is as follows:

$$com(v, r) = g_1^{v_1} \cdot g_2^{v_2} \cdot \ldots \cdot g_N^{v_N} \cdot h^r$$

Similarly, the Pedersen vector commitment over the elliptic curve is given as follows:

$$com(v, r) = v_1 G_1 + v_2 G_2 + \ldots + v_N G_n + rH$$

For the Pedersen commitment, the most important element is the choice of the group G. In this E-voting system proposal, we chose to use the time-tested Montgomery elliptic curve [20], specifically the Curve25519 curve.

To speed up the computation from the x coordinate of a point from coordinates of two other points over the Curve25519 Montgomery curve, we used the Okeya–Sakurai y-recovery algorithm [21].

**Theorem 1.** *[Okeya–Sakurai y-recovery algorithm] Let $P = (x, y)$, $P_1 = (x_1, y_1)$, $P_2 = (x_2, y_2)$ be points over an elliptic curve of the Montgomery type. If $P_2 = P_1 + P$, $y \neq 0$, A, B are parameters of the Montgomery elliptic curve and $B \cdot (A^2 - 4) \neq 0$, then the following equation holds:*

$$y_1 = \frac{(x_1 x + 1)(x_1 + x + 2A) - 2A - (x_1 - x)^2 x_2}{2By}$$

### 3.3.3. zk-SNARKs

There are currently many algorithms for zk-SNARKs. Ronald Mannak found through comparing some zk-SNARKs algorithms that the Groth16 algorithm outperformed other algorithms in terms of proving data size and validation speed [22,23]. Compared with other zero-knowledge proof systems, it has higher efficiency and smaller proof size in practical applications.

- *Setup*: The setup phase mainly exists to generate the verification key and proof key for the algorithmic process, where the verification key is used to verify the correctness.
- *Prove*: In the proof phase, the prover generates a zero-knowledge proof based on the parameters and the given computational task. This proof shows that the prover knows a secret input that satisfies a particular relation, but does not reveal any information about this input. The proof process produces a compact proof that can be quickly checked by a verifier.
- *Verification*: In the verification phase, the verifier checks the correctness of the computation using public parameters and proofs generated by the prover. The verifier does not need to know the secret inputs used by the prover, nor does it need to perform the actual computation. The verification process is efficient and scalable.

### 3.3.4. Face Recognition Based on Convolutional Neural Networks

Convolutional neural networks (CNNs) have become a research hotspot in many scientific fields, especially in the field of pattern classification, in which it is more widely used because the network avoids the complex pre-processing of images and can be directly input to the original image. Generally, the basic structure of a CNN consists of two layers, one of which is a feature extraction layer, where the input of each neuron is connected to the local receptive domain of the previous layer and extracts the features of that local area. Once this local feature is extracted, the positional relationship between it and other features is also determined. The second is the feature mapping layer. Each computational layer of the network consists of multiple feature mappings; each feature mapping is a plane, and the weights of all neurons on the plane are equal. The feature mapping structure uses a sigmoid function with a small influence function kernel as the activation function of the convolutional network, making the feature mapping displacement invariant.

Since the feature detection layer of CNN learns through training data, the feature extraction of displays is avoided, based on CNNs, which implicitly learn from the training data. In addition to the neurons on the same feature mapping surface having the same weights, the network can be learned in parallel, which is a major advantage of CNNs. CNNs, with their special structure that involves local weight sharing, have a unique superiority in speech recognition and image processing; their layout is closer to that of the actual biological

neural network. Weight sharing reduces the complexity of the network; in particular, multi-dimensional input vector images can be directly input into the network. This feature avoids the complexity of data reconstruction in the process of feature extraction and classification. The general technical lines of face recognition based on convolutional neural networks are depicted as Figure 2.

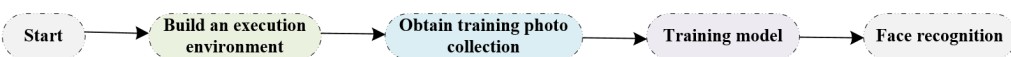

**Figure 2.** Technical routes of the face recognition.

## 4. System Model and Security Requirements

In this section, we provide a system model and its security requirements, and present a model of our proposed electronic voting system.

### 4.1. System Model

There are five main entities in our E-voting system, namely the AI Engine, voters, the teller, the organization, and the bulletin board. Our system model is presented in Figure 3.

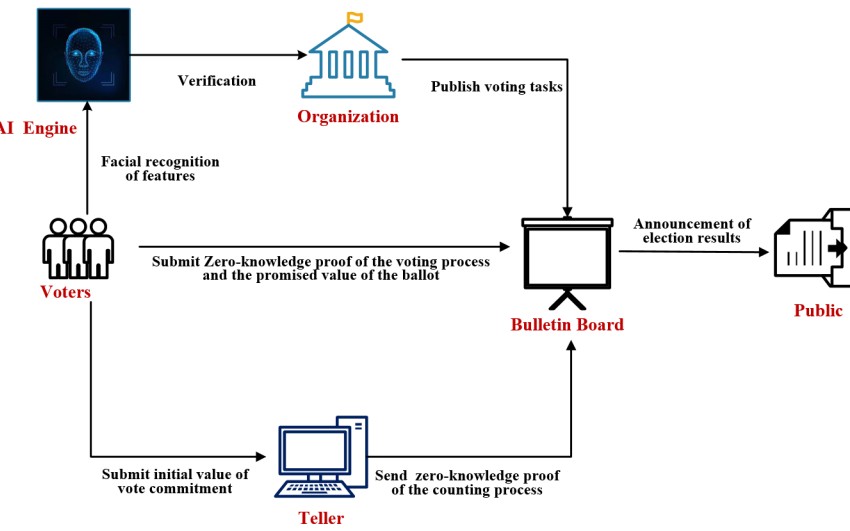

**Figure 3.** System model of our constructed E-voting system.

1. *Voters*: Voters $V_i$ ($i = 1, 2, \ldots, n_v$) are eligible to vote in an election. Generally speaking, voters must fulfill certain qualifications and conditions to participate in voting, such as age, nationality, identity, and so on. In democratic countries, voters are usually registered and managed by the government or an independent organization to ensure the authenticity and legitimacy of voter identity. The voting behavior of voters has a significant impact on the outcome of an election, so it is necessary to safeguard the rights and interests of voters during the election process, while ensuring that the election is fair, open, and transparent.

2. *Teller*: A teller $T_i$ for $i = 1, 2, \ldots, n_v$, known as a tallying officer, is a person responsible for counting and recording the votes cast in an election. During an election, after voters have cast their ballots, tellers are required to count and tally the ballots to determine the number of votes cast for each candidate. The role of the tellers is to ensure the accuracy and fairness of the election results, depending on their operations and judgments. In some countries, counting agents are employed by government or independent organizations to ensure the objectivity and fairness of the election results.
   In some places, counting agents are also required to comply with certain requirements to ensure the legitimacy and transparency of the electoral process.

3. *Organization*: The sponsoring organization is usually the national government, company directors, etc. In an election scheme, the sponsoring organization should be

authoritative to ensure fairness, objectivity, and transparency and openness in the election process. The organization can ensure the verifiability and credibility of the election process through allowing results to be audited and verified by supervisors or other independent bodies.

4. *Bulletin board*: A bulletin board in the context of this E-voting system means a publicly verifiable bulletin board. Voters, counting agents, and the organization can communicate information to form a bulletin board.

Our construction has many symbols listed in Table 1.

**Table 1.** Parameters of our construction.

| Notation | Descriptions |
|---|---|
| $V_i$ | Voters |
| $T_i$ | Tally statistics |
| $pk, sk$ | Public and private keys |
| $n_{choice}$ | Number of candidates for this election |
| $C$ | Voting options for this election |
| $f_{res}$ | Election function of ballot papers |
| $n_{choice} = n_{tuples} \cdot N$ | Split the voting into $n_{choice}$ tuples |
| $(m^{i,1}, \ldots, m^{i,n_{choices}})$ | Original ballot paper of voters |
| $(t_k^{i,1}, \ldots, t_k^{i,n_{choices}})$ | Decomposition tuple for raw voter ballots |
| $c_k^{i,l} \leftarrow com(t_k^{i,l}; r_k^{i,l})$ | Voters create a pledge for each tuple |
| $c^{\perp,l} \leftarrow com(t^{\perp,l}; r^{\perp,l})$ | Counters doing pledge summaries |
| $Z_{ballot}^i$ | Proof of Groth 16 |
| $Enc(\cdot)$ | Encryption |
| $Dec(\cdot)$ | Decryption |
| $T := (m^{\perp,1}, \ldots, m^{\perp,n_{choices}})$ | Counting of all ballots |
| $res$ | Election results |

### 4.2. Security Requirements

For voting systems based on machine learning, the following security requirements must be satisfied:

- *Traceability of counting results*. Due to the definition of concealed public counting, the teller can count the votes internally but cannot publish the complete counting results (only the final election results). Therefore, in our voting system, the honesty of the counting agents should be strictly checked, and the counting results should be traceable.
- *Privacy*. In this E-voting system, the ballot information must not be leaked, as this may indirectly lead to the leakage of the voters' personal information or a decline in the authority of the final winner.
- *Verifiability*. The main purpose of a certain voting system for elections is to give the public confidence in the results of said elections. The verifiability of the voting system has a bearing on the level of trust and voter acceptance of the electoral process. Failure to ensure verifiability in the voting process will significantly reduce public acceptance of the results. Once the public questions or mistrusts the results of the election, the election will be meaningless. Therefore, the design of this system should strictly ensure the verifiability of the voting system.

### 4.3. Threat Modelling

The threat model of our proposed E-voting system consists of the following:

- *Ballot content altered*: Attackers may tamper with ballots in a variety of ways, including hacking, malware, hardware attacks, and so on. By tampering with the ballot paper, the attacker can change the result of the election and thus achieve their improper purpose.
- *Voter identity theft*: Attackers may steal voters' identities and impersonate them to vote. By stealing a voter's identity, an attacker can influence the outcome of an election and control the process.

- *Compromised voting information*: Attackers may obtain voting information through hacking, social engineering, etc., and use it for improper purposes. By leaking voting information, attackers may be able to influence voters' behavior, thereby affecting the outcome of the election.

Therefore, it is necessary to make targeted improvements to the threat model so as to ensure the security of the voting system and the personal privacy of voters and to protect the legal compliance of voters' right to vote.

## 5. Our Construction

In this section, Pedersen commitments are used to convert a vote into a commitment. In turn, zk-SNARKS are used to validate the Pedersen commitments of the ballots to ensure the correctness and security of the commitment of the ballots. In addition, for the case in which there is more than one teller, zk-SNARKS can generate a proof for each teller. These proofs can be verified to ensure the accuracy of counts. Throughout the process, facial recognition function based on convolutional neural networks will effectively provide the correct identification of voters, further enhancing the security of the system solution.

1.  **Initialization.** The initialization phase mostly involves completing some preparatory work.

    - *Organization*: At the initialization stage, the host organization should complete the work of formulating election rules and procedures, registering voter information, publicizing election information, and establishing a legitimate monitoring mechanism to ensure that the election process is fair, open and transparent and that the voters' right to vote and their interests are safeguarded. At the same time, the host organization also needs to stipulate the various other parameters of the voting and counting process. In addition, the host organization should obtain the correct facial images of voters in advance, and accordingly use them for training.
    - *Bulletin board*: During the initialization phase, the host organization should publish the schedule time of the election, a list of legal voters, the format of the ballot paper, and other information on the bulletin board for the participants in the election process to view. Moreover, the host organization needs to publish the various parameters for the creation of the Groth16 certificate on the bulletin board.
    - *Voter*: At the initialization stage, the main tasks of the voter are to determine his/her eligibility to vote according to what is stated on the bulletin board, to determine the start and end time of the voting, to understand and learn about the rules and procedures of the election, and to study the candidates or options. At the same time, the voter should provide the responsible organization with a correct image of their face.
    - *Counters*: In the initialization phase, the task of each counter is to generate his/her own public and private key according to the selected public key encryption scheme and publish the public key, etc., on the bulletin board.

2.  **Voting**. In this phase, the main process is as follows.

    - The organization publishes the number of candidates for this election $n_{choice}$ and the voting options (choice space) for this election C. A deterministic polynomial time function is used to calculate the final election result based on the votes $f_{res}$. In order to facilitate the use of commitment vectors in subsequent calculations, the organization needs to make an advance provision:

    $$n_{choice} = n_{tuples} \cdot N$$

    The reason for this specification is to decompose the choice space by splitting $V \in C$ into $n_{tuples}$ tuples, each of size $N$. If $n_{tuples}$ cannot be decomposed by integrating $N$, options can be added artificially (e.g., constant to a value such as 0).

- For each teller $T_k$, the original votes $(m^{i,1}, \ldots, m^{i,n_{choices}})$ of voter $V_i$ are decomposed into $n_{tuples}$ tuples $(t_k^{i,1}, \ldots, t_k^{i,n_{choices}})$. The voter creates a pledge for each tuple: $c_k^{i,l} \leftarrow com(t_k^{i,l}; r_k^{i,l})$. Since Pedersen vector commitments are additively homomorphic, commitments to the original votes of the voters $V_i$ can be obtained by combining the commitments of all tuples.

- Voters create a Groth16 proof for their vote $Z_{ballot}^i$. $Z_{ballot}^i$ can be used to prove voters' promises and their personal information, and it can prove that the promises $(c^{i,1}, \ldots, c^{i,n_{choices}})$ were submitted to the ballot $m_i \in C$.

- After the voter has fulfilled correct face recognition, the public key $Pk_k$ corresponding to each teller $T_k$ is used to send the initial value $(c_k^{i,1}, \ldots, c_k^{i,n_{choices}})$ of the key to the corresponding counting agent:

$$e_k^i \leftarrow Enc(pk_k, ((t_k^{i,1}, \ldots, t_k^{i,n_{choices}}), (r_k^{i,1}, \ldots, r_k^{i,n_{tuples}}))).$$

- After the voter performs correct face recognition, he/she submits the ballot paper to the bulletin board $b^i = (i, (c_k^{i,1})_{l,k}, Z_{ballot}^i, (e_k^i)_k)$. After receiving the voter's data, the bulletin board conducts verification and checking operations, which mainly include:

  (a) Checking whether the voter is eligible to vote;
  (b) Checking for duplicate submissions of ballot papers by voters;
  (c) Checking the validity of the voters, using a Groth16 certificate.

  If all checks are successful, the bulletin board counts $b_i$ on the ballot $b$ and the public list $B$ (which only the voter $V_i$ can see for himself).

- Afterwards, each teller decrypts the $e_k^i$ decryption:

$$((t_k^{i,1}, \ldots, t_k^{i,n_{tuples}}), (r_k^{i,1}, \ldots, r_k^{i,n_{tuples}})) \leftarrow Dec(sk_k, e_k^i)$$

  Each counter $T_k$ then checks whether each initial commitment tuple $(t_k^{i,l}, r_k^{i,l})$ corresponds to the corresponding $c_k^{i,l}$. If it does not correspond, the corresponding teller $T_k$ generates a zero-knowledge proof against the checking process to send to the bulletin board in order to verify that the $e_k^i$ is invalid. The bulletin board deletes the corresponding ballot, and the corresponding ballot will not be counted in the counting phase, wherein $T_k$ checks whether the initial value provided by the voter corresponds to the generated promise value. This process does not reveal the voter's choice because the promise value is only a representation of the ciphertext and does not contain specific plaintext information.

3. **Counting**. Once all voters have submitted valid ballot papers, the counting stage will begin. The main processes in the phase are as follows.

- In the vote counting phase, when $1 \leq l \leq n_{tuples}$, everyone can homomorphically count the promises of the public list B:

$$c^{\perp,l} \leftarrow \sum_{k=1}^{n_t} \sum_{i=1}^{n_v} c_k^{i,l}$$

- When $1 \leq l \leq n_{tuples}$, the counters can compute the corresponding opening of the commitment in an internal homomorphic aggregation:

  - When $1 \leq l \leq n_{tuples}$, each teller counts internally:

$$t_k^{\perp,l} \leftarrow \sum_{i=1}^{n_v} t_k^{i,l}$$

$$r_k^{\perp,l} \leftarrow \sum_{i=1}^{n_v} r_k^{i,l}$$

- Teller $T_k$ can be shared between $(t_k^{\perp,l})_l$ and $(r_k^{\perp,l})_l$, therefore, $1 \leq l \leq n_{tuples}$ can be calculated:

$$t^{\perp,l} \leftarrow \sum_{k=1}^{n_t} t_k^{i,l}$$

$$r^{\perp,l} \leftarrow \sum_{k=1}^{n_t} r_k^{i,l}$$

After this step, all counters can calculate the count of all votes cast:

$$T := (m^{\perp,1}, \ldots, m^{\perp,n_{choices}})$$

where $m^{(\perp,j)}$ is the total number of votes for the candidate $j$ in the election process.

- The final step is completed by the designated tellers.
  - The teller calculates the results of the election based on the votes counted.

$$res \leftarrow f_{res}(T)$$

  - The teller creates a Groth16 proof based on $c^{\perp,1}, \ldots, c^{\perp,n_{tuples}}$, and $res$ creates a Groth16 proof and posts it on the bulletin board along with $res$ to prove that the $res$ was indeed generated by the $res \leftarrow f_{res}(T)$. Specifically, $c^{\perp,1}, \ldots, c^{\perp,n_{tuples}}$, and $res$ are used as public inputs, and the initial value of the voter's promise is used as a secret input to prove that the initial value corresponds to the promise; this means the proof $c^{\perp,1}, \ldots, c^{\perp,n_{tuples}}$ is $T := (m^{\perp,1}, \ldots, m^{\perp,n_{choices}})$. $res$ is generated by $res \leftarrow f_{res}(T)$. Since Groth16 proofs are non-interactive zero-knowledge proofs, others (e.g., supervisors) can verify that the knowledge is correct without having the knowledge $T := (m^{\perp,1}, \ldots, m^{\perp,n_{choices}})$.

## 6. Security Analysis

As our E-voting system is based on hidden public counting, the Pedersen commitment, the Groth16 algorithm, etc., the security of our E-voting system is analyzed in three aspects. In this system, the honesty of the counting agents is based on the concept of hidden public counting, while the Pedersen commitment and Groth16 algorithm ensure the privacy and verifiability of this system. For this electronic voting system, if there are any issues in the voting, counting, and supervision processes, the results of this election will not be recognized.

### 6.1. Privacy

**Theorem 2.** *If the discrete logarithm problem is hard to solve and the Pedersen commitment is secure, our constructed system has boundedness.*

**Proof.** The Pedersen commitment has the property of a hidden message $v$, i.e., an attacker cannot recover the value of $v$ from the commitment value *com*, and also has the property of a bound message $v$, i.e., an attacker cannot change the commitment value *com* by changing the random number $r$.

Specifically, the process of generating commitment values involves the computation of the discrete logarithm problem, in which, given a large prime $p$, a generator $g$ and a number $y$, we may find an integer $x$ such that $g^x = y \mod p$. This problem is computationally intractable because there is no efficient algorithm to solve it. That is, the discrete logarithm problem is difficult based on the discrete logarithm problem, and Pedersen's commitment has boundedness.  $\square$

**Theorem 3.** *If the Pedersen commitment is hidden, then this system has the property of hiddenness based on a mathematical hard problem.*

**Proof.** The hiddenness of the Pedersen commitment means that the data are invisible and cannot be observed or inferred by others. Specifically, due to the introduction of the blind factor $r$ in the commitment, different commitments $c$ can be generated for the same $v$; even if sensitive privacy data $v$ remain unchanged, the final commitment $c$ will change with the change in $r$, thus providing the invisibility and security of the information theory. It can be seen that Pedersen's commitment generation methods are similar to algorithms such as encryption and signature. However, as a promise of cryptography, the emphasis is on "promise", and does not provide a decryption algorithm. This means that if only $r$ is known, it cannot effectively calculate private data value $v$. Pedersen commitments achieve data hiding by encrypting and obfuscating the data, which protects the data in the commitment from leakage. After committing plaintext using the Pedersen commitment, an attacker cannot infer any original data since the data-hiding nature of the Pedersen commitment is based on the discrete logarithm problem. □

**Theorem 4.** *If Groth16 is secure and has properties of completeness, soundness and zero knowledge, then our construction has completeness, soundness and zero knowledge.*

**Proof.** It is easy to obtain this theorem according to our construction [23,24]. The security of the Groth16 algorithm relies on the security of bilinear pairs in addition to being based on the discrete logarithm problem. The security of bilinear pairs depends on the difficulty of the Bilinear Diffie–Hellman Problem (BDHP). The Bilinear Diffie–Hellman Problem (BDHP) is the problem of solving $e(R, S)$, given $P, Q, e(P, Q), R, S, e(R, Q)$, and $e(P, S)$ in a bilinear mapping. Although the BDHP can be solved theoretically, there is no known efficient algorithm that can be used to solve the problem efficiently. Thus, the security of the Groth16 algorithm can be guaranteed as long as these two problems are still considered difficult. Based on bilinear pairs and elliptic curve cryptography, the Groth16 algorithm guarantees completeness, reliability and zero knowledge.

- Completeness: The Groth16 algorithm guarantees completeness, i.e., if a statement is true, then there exists proof that it is true. This means that if legitimate proof exists, then the statement must be correct. The Groth16 algorithm achieves completeness based on bilinear pairs. For this system, the zero-knowledge proof of the voting process submitted by the voter $V_i$ and the zero-knowledge proof of the counting process submitted by the teller $T_i$ are able to guarantee that the voting and counting processes are true.
- Soundness: If a statement is wrong, there is no proof that can trick the verifier into accepting the wrong statement. The Groth16 algorithm achieves reliability using a zero-knowledge proof system based on bilinear pairs on elliptic curves. For this system, if the voter $V_i$ submits zero-knowledge proof of the voting process, and the zero-knowledge proof of the counting process submitted by the teller $T_i$ is illegal (after an effective attack), the vote will not pass the checking and acceptance stages of this E-voting system.
- Zero-knowledge: The Groth16 algorithm guarantees zero knowledge, i.e., the prover can prove a proof that it knows the statement without revealing any information about said proof. The Groth16 algorithm uses a zero-knowledge proof system in which the prover can generate an encrypted proof that allows the verifier to verify the correctness of the proof but cannot obtain any information about the prover's knowledge from this proof. Within this system, if a voter $V_i$ submits a zero-knowledge proof of the voting process, and teller $T_i$ submits a zero-knowledge proof of the tallying process but does not disclose any valid information to the outside world, the prover can then prove that he/she knows the proof of a statement without disclosing any information about the proof.

  □

## 6.2. Hidden Counting

We propose herein a verifiable E-voting system solution based on the concept that public vote counting can be hidden. The simple concept of hidden counting cannot guarantee verifiability while protecting voters' privacy.

**Fully hidden counting**. Complete counting concealment means that only the actual election results are made public, without any intermediate counting results. Neither internal counters nor external voters can learn of any intermediate results. This approach usually requires the use of heavyweight cryptographic techniques, such as secure multiparty computation (MPC). The concept of complete vote count concealment limits its applicability to elections with a certain number of candidates; it can only be used for simple voting methods and a small number of candidates.

**Partial hidden counting**. Partial vote count concealment, which involves hiding only the counts that lead to a particular problem, allows for a more efficient way of handling complex elections. However, disclosing a part of the ballot still reveals information, such as the order of the candidates, so certain problems remain.

**Public hidden counting**. Within the concept of public hidden counting, the teller can count the votes internally but does not publish the complete counting results, only the final election results. This mechanism exists to protect voters' privacy. It may be relatively more effective in protecting the privacy of voters compared with fully hidden counting and partially hidden counting and can be achieved using zk-SNARKs and some cryptographic techniques.

## 6.3. Verifiability

The verifiability of this E-voting system is based on a verifiability framework used in mainstream E-voting protocols [25]. This framework assumes the existence of a "virtual" entity, called a "judge J", whose role is to accept or reject the operation of the election protocol. In an actual election, the judge's procedures can be carried out by any party, including external observers, or even the voters themselves. Judges use only public information as the input (e.g., zero-knowledge certificates) for performing certain checks. In the context of E-voting, in order to achieve verifiability, the judge should only consent to proceed if "the declared election result corresponds to the voters' actual choice", i.e., in an E-voting system, the result of the vote should correspond to the voters' actual choice. If the result does not correspond to the actual choice of the voters, then the judge will reject the operation of the electoral protocol, i.e., the result of the election.

The framework defines a goal $Y$, which is an election protocol considered to be valid only if its results match the actual choices made by voters. The role of this framework is to ensure that the results of votes cast within in an electronic voting system are not tampered with through fraudulent behavior, thereby safeguarding voters' right to vote and the accuracy of the election results. Therefore, based on the above framework, two assumptions of this E-voting system are presented herein.

**Assumption 1.** *The public key encryption and decryption scheme are correct, the commitment scheme is homomorphic, and the zero-knowledge proof is sound.*

**Assumption 2.** *Judge J and the bulletin board are honest.*

The verifiability of our system is proposed based on the above assumptions. The goal $Y$ is validated by the judge $J$ in the election protocol. The idea behind the definition is simple: the judge $J$ accepts the election only if the objective $Y$ is satisfied, so that the published election results correspond to the actual choices of the voters. In this E-voting system, the judge reads all the data from the bulletin board and accepts the protocol; that is, they receive the results of the election if and only if all zero-knowledge proofs on the bulletin board are valid.

We compared our system with alternative voting systems in terms of security as shown in Table 2.

**Table 2.** Comparison of four related works.

| Electronic Voting | Counting Hidden | Verifiability | Identity Privacy | Intelligent | Coercion resistance |
|---|---|---|---|---|---|
| Canard [17] | √ | × | × | × | × |
| Ramchen [9] | × | × | × | × | √ |
| Ordinos [2,10] | √ | √ | × | × | × |
| Our system | √ | √ | √ | √ | √ |

Among them, √ represents possessing the property, and × represents not possessing the property

The Ordinos electronic voting system is the first provably secure, verifiable, fully counted, hidden electronic voting system that has been widely used in various election scenarios and as part of various voting methods. Ordinos is based on the concept of complete vote counting and hiding, which is practical for elections involving fairly simple voting methods (single choice or multiple choice) and a limited number of options (but a large number of voters). That said, this system is based on the concept of open counting and hiding, and is practical for elections involving very complex voting methods (e.g., majority rule and instant runoff), and is quite effective in dealing with a large number of options. Compared with public hidden vote counting and completely hiding vote counting, completely hidden vote counting provides better privacy protection, while the public hiding vote counting systems can provide better efficiency. In addition, the Ordinos electronic voting system is implemented using secure multi-party computation (MPC) technology, and this electronic voting system uses homomorphic commitments and concise and efficient zero-knowledge proofs that are more lightweight than secure multi-party computation.

Canard et al. [17] designed and implemented an electronic voting system for majority decision voting in order to implement a method based on the concept of complete vote hiding. However, compared with the electronic voting system presented herein, it took a relatively longer time to calculate the same number of tasks. This electronic voting system can not only implement majority decision voting but is also more efficient in terms of calculation efficiency. In addition, the electronic voting system designed by Canard et al. [17] has still not been the subject of a security analysis or verifiability discussion.

Ramchen et al. designed an electronic voting system based on the concept of partial vote hiding, which has played a major role in preventing the exposure of private information during elections. However, through this system, voters can learn some intermediate information during the election process, which may cause embarrassment to some candidates who received a low number of votes. Our electronic voting system avoids such pitfalls because the public are only given the final election results.

This electronic voting system also achieves coercion resistance. Coercion resistance refers to an important concept in the design of electronic voting systems that aims to ensure that voters can freely and privately express their votes without any form of coercion or threat. In a coercive environment, voters may face threats from others, such as employers, family members, government agencies, or other potentially malicious parties. The goal of coercion resistance is to protect voters from external pressure and ensure that their votes are authentic and uninfluenced. In this electronic voting system, the realization of coercion resistance is mainly based on the following aspects: (1) Privacy (ballot secrecy): Protect voters' voting information from being leaked to prevent anyone from tracking or identifying their votes. In this electronic voting system, voters' voting information cannot be tracked or checked. (2) Verifiability: Provide a mechanism that allows voters to verify whether their votes were correctly recorded and calculated without revealing the content of their actual votes. In this electronic voting system, the verifiability of voting is achieved by using commitment and zero-knowledge proof methods. (3) Non-coercibility: The system is designed to ensure that if forced, voters cannot provide the evidence required by the

coercer to prove their voting method. In this electronic voting system, cryptography-based technology achieves this. (4) Selective disclosure: Allow voters to selectively disclose their voting information in order to prove their vote when verification is required, rather than being forced to disclose it when they do not want to. In this electronic voting system, due to the use of zero-knowledge proof and commitment technologies, voters can selectively disclose their voting information to prove their votes when verification is required. Based on the above properties, this paper also achieves coercion resistance.

### 6.4. Facial Recognition Security Proof

Due to the references [26,27], we use privacy-enhancing face biometrics to protect the privacy of the person in our construction.

Privacy of the person is related to the integrity of a person's body. Threats to this type of privacy include physical intrusions, such as torture or compulsory medical treatment, immunization, and also forced biometric measurement. Privacy of personal behavior (or media privacy) is concerned with sensitive behavioral information, such as political activities, sexual habits, or religious practices, but also with the personal space (private or public) needed to facilitate such behavior. Privacy of personal communications (or interception privacy) is associated with the ability to communicate freely using various means (e.g., verbal, written, gestured, and electronic) without being monitored by third parties. Privacy of personal data (or data privacy) is related to the general availability of personal data and the ability to execute control over one's personal data and their use by third parties. This type of privacy is (often jointly with privacy of personal communications) also referred to as information privacy.

## 7. Performance Evaluation

This chapter describes the implementation and validation of multiple voting methods based on this E-voting system solution, alongside some analysis and discussion of the test results produced using these methods. Our E-voting system's implementation was primarily carried out using Windows 11 with an Intel(R) Core(TM) i5-8300H CPU and 8.00 GB of onboard RAM.

### 7.1. Pedersen Commitment

The implementation of the Pedersen commitment in this E-voting system is based on the Montgomery curve. In its practical implementation, a list of voters' ballots is used as the original information vector $v_1, v_2, \ldots, v_N$, and the points on the Montgomery curve are committed as $g_i$ and $h$ :

$$\mathrm{com}(\mathbf{v}, r) = g_1 v_1 + g_2 v_2 + \ldots g_N v_N + hr$$

Based on the above parameters, Table 3 shows some of the test results of the Pedersen commitment in this E-voting system.

**Table 3.** Related tests for Pedersen commitment.

| Module Test | Time (s) |
|---|---|
| Pedersen commitment of the Montgomery curve | 0.002 |
| Pedersen commitment when $h$ is incomplete | 0.005 |
| Pedersen commitment when $g$ is incomplete | 0.004 |
| Pedersen commitment when both $g$ and $h$ are incomplete | 0.003 |
| Pedersen commitment when $h$ is the origin | 0.003 |
| Montgomery curve Pedersen vector commitment (double random factors) | 0.005 |
| Montgomery curve Pedersen vector commitment (single random factor) | 0.006 |

As shown above, the tests of the Pedersen commitments for multiple scenarios have a commitment time of no more than 0.01 s for single or multiple data.

### 7.2. Time Testing for Multiple Voting Methods

Single- and multiple-choice voting methods are common voting methods used primarily to count the top $n$ options that receive the most votes. The single-choice voting method is a voting method in which each voter can select only one option. In single-choice voting, the number of votes each option receives is relative, and the ultimate winner is the option that receives the most votes. The multiple-choice voting method, on the other hand, is a voting method in which each voter can choose more than one option. In multiple-choice voting, the number of votes each option receives is absolute, i.e., it is as simply as many votes as it receives. The ultimate winner is either the option that receives the most votes or the option that receives a particular percentage of the total votes (e.g., 50 percent).

This voting method can be used to calculate the $n$ top candidates during the actual election process. In the actual testing process, for example, in an election with 5 candidates and 1000 ballot data, the $n$ top options that received the most votes are calculated separately ($n = 1, \ldots, 6$). The computational time in different cases is given as follows.

Single-choice voting methods are simple to understand and easy to implement and calculate but may not reflect the diversity and complexity of voters' opinions. Multiple-choice voting methods can better reflect the diversity and complexity of voters' opinions but may require more complex calculations and have greater technical requirements. Therefore, the choice of the voting method needs to be made on a case-by-case basis. As shown in Table 4, when using this E-voting system for single- and multiple-choice voting, we found the calculation time to be very short (no more than 0.5s), which can enhance efficiency in practical elections.

**Table 4.** Time consumption for single- and multiple-choice voting methods.

| Test Module Role | Calculation Time (s) |
| --- | --- |
| The option with the most votes | 0.245 |
| The top 2 options that received the most votes | 0.216 |
| The top 3 options that received the most votes | 0.237 |
| The top 4 options that received the most votes | 0.260 |
| The top 5 options that received the most votes | 0.210 |

### 7.3. Time Testing for Majority Judgment Voting

Majority judgment voting is a voting method used to select a candidate or option, in which a choice is given to voters; voters are asked to rank or rate each option, and the median is calculated for each option. All medians are then compared to all other medians, and the option with the highest median is chosen as the winner. If no such unique winner exists, this voting method is executed in multiple rounds. During each round, the best median score is identified. All candidates with poorer current median scores are eliminated.

Then, for each remaining candidate, one vote is removed from that median score. In the next round of voting, the median score is updated accordingly, as single votes are canceled. This process is repeated until only one candidate remains to be declared the winner of the election.

For the actual test, suppose six people are discussing where to hold a social gathering, and they consider three alternatives:

- Option 1: Meet at a bar;
- Option 2: Host a picnic in an outdoor park;
- Option 3: Dine in an indoor restaurant.

To aid decision making, majority judgment voting can be used to determine which alternative is most popular. Each participant can rate each alternative and store these voting ratings in a list, each representing the voting rating of an alternative. The voting grades for the three alternatives are as follows:

- Option 1: [1, 2, 2, 1];

- Option 2: [2, 1, 3, 0];
- Option 3: [0, 6, 0, 0].

Ultimately, for the different numbers of winning scenarios, the time taken for the majority judgment method to be executed was as follows.

As shown in Table 5, when using this E-voting system solution for majority judgment voting, we found the time needed for computation to be very minimal (generally no more than 0.01 s), meaning the efficiency of actual elections can be enhanced.

**Table 5.** Time cost of majority judgment voting methods.

| Module | Time (s) |
| --- | --- |
| The sole winner of the majority judgment | 0.006 |
| The first two winners of majority judgment | 0.003 |

### 7.4. Time Testing for Groth 16

After deploying Libsnark, we assessed the benchmarks of various proof systems on an R1CS using the example of 1000 constraints and 1000 variables, of which 10 were input variables. The benchmarks were obtained using a 2.60 GHz Intel Core i7-9750H CPU in single-thread mode, using the BN128 curve. Finally, we obtained a zero-knowledge proof that it only takes 20.63 s to complete Groth16.

### 7.5. Time Testing for Face Recognition Based on a Convolutional Neural Network

The choice of other people's image sets, i.e., datasets, affects the results of model training and relates to the construction of neural network models; the choice of appropriate datasets is a guarantee of the accuracy of neural network models. The size of each color image is $250 \times 250$, and the LFW dataset is mainly used to test accuracy in face recognition, from which 6000 pairs of faces are randomly selected to form face recognition image pairs (among which 3000 pairs belong to the same person with two face photos, and 3000 pairs belong to different people with one face photo each). During the test, LFW produces a pair of photos and asks the system in the test if the two photos are of the same person, and the system gives a "yes" or "no" answer. The face recognition accuracy is calculated using the ratio of the system's answers to the true answers across 6000 pairs of face test results. This set is widely used to evaluate face recognition performance.

In the actual test, we mainly utilized the powerful function library of Python to first train the system with the LFW dataset and the correct photoset of the voters. Then, the online face recognition of voters could be performed in real time with no less than ninety-eight percent recognition accuracy. The time taken up by the whole process is shown in Table 6.

**Table 6.** Face recognition testing.

| Test Module Role | Calculation Time (s) |
| --- | --- |
| Camera recognition time | 0.13 s–0.15 s |
| Model training time | 5 min 30 s |
| Photo collection time | With a high-speed camera, 500 frames/s, takes 20 s |

Finally, we compared our construction with related works; we conducted a comparison test of our E-voting system and the Python source code of the Ordinos E-voting system in Github, and the results of the test are shown in the figure below. As shown in Figure 4, our scheme has advantages in terms of performance and function.

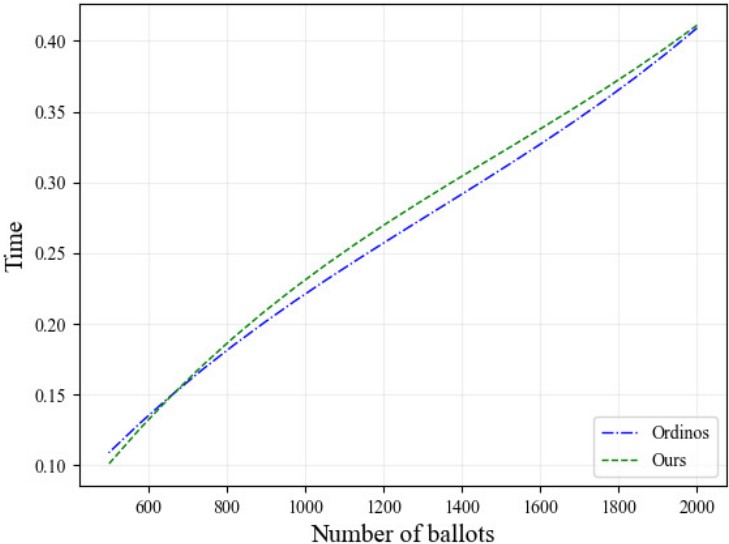

**Figure 4.** Comparative analysis.

*7.6. Total Performance Evaluation*

In this subsection, we describe an evaluation of the performance of our construction, which includes the execution time needed for each phase and some construction modules, as shown in the Figure 5.

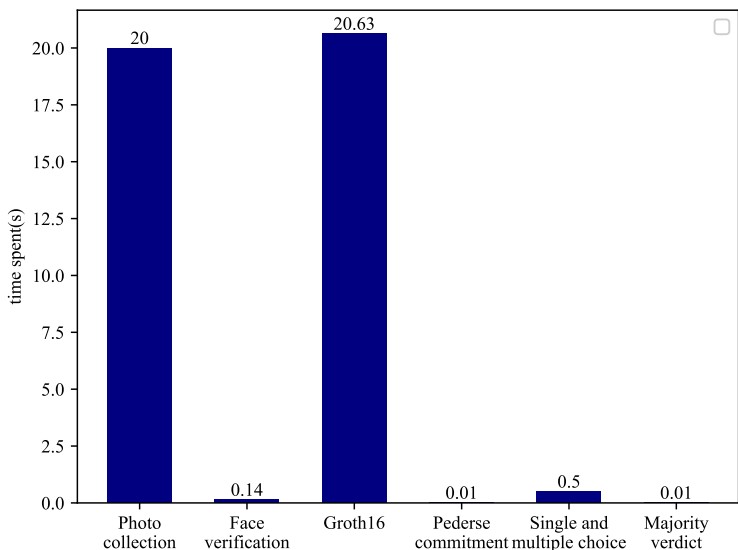

**Figure 5.** Time cost of our construction.

## 8. Conclusions

In this paper, we used some cryptographic primitives, including a commitment scheme and a zero-knowledge proof, combined with face recognition technology based on a CNN, to establish a publicly verifiable E-voting system. Through experimental testing and security analysis, our construction meets some security requirements and has some advantages compared with related electronic voting systems. In the future, we will consider how our scheme may be improved for future related applications.

Our system can protect the identity of each user through the user's biometric information because biometric technology has some unique advantages and security features. First, it cannot be forged. Facial features are unique to each person and difficult to forge or impersonate. This means that only legitimate voters can vote using their own biometric

information, thereby preventing voter fraud. Secondly, it has high-precision features. The facial recognition technology used in our electronic voting system is highly accurate and can precisely identify individuals. This helps to ensure that each voter can only vote once, thus reducing the possibility of double voting. Finally, this technology is unique. Unlike traditional voting systems, facial feature information does not require people to memorize passwords or carry various identity documents, making voting more convenient and easier to do, while also reducing the probability of harm caused by leaking passwords or losing personal identity documents. Therefore, our system prevents voters from impersonating others in order to vote because only their biometric information can be used to vote.

**Author Contributions:** Conceptualization, J.L. and T.H.; methodology, J.L., T.H.; software, M.T.; formal analysis, T.H.; investigation, B.T., W.H., Y.Y.; writing—original draft preparation, J.L., T.H. and M.T.; writing—review and editing, J.L., T.H. and M.T.; supervision, B.T., W.H., Y.Y. All authors have read and agreed to the published version of the manuscript.

**Funding:** This research was funded by the National Natural Science Foundation of China (61872239, 61872229, 62272389, U19B2021, U20B2064), Natural Science Basic Research Plan in Shaanxi Province of China (2021ZDLGY06-04, 2020ZDLGY09-04, 2021ZDLGY05-01), Shenzhen Fundamental Research Program (20210317191843003).

**Data Availability Statement:** Data will be made available on request.

**Conflicts of Interest:** There exists no conflicts of interest.

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
