# Peer review of "RETRACTED: A Publicly Verifiable E-Voting System Based on Biometrics"

_cryptography, doi:10.3390/cryptography7040062_

Round 1
Reviewer 1 Report
Comments and Suggestions for Authors
(1) Title: Machine learning is missleading: The revision may chance to Face recognition or Biometrics
(2) The comparison Table-2 is not fair: Ref-[24] discusses Coercion resistance of e-voting system, whereas the authors’ submission does not. Revise Table-2 for being Fair. Does the authors’ proposal achieve coercion resistance?
(3) Discussion on Related works with references is not enough. Add more related works on e-voting with biometrics including Face-recognition.
E.g
https://dl.acm.org/doi/10.1016/j.tcs.2022.05.005
https://www.sciencedirect.com/science/article/abs/pii/S0304397522002869
The authors shall review more related papers like above by authors themselves.
(4) Discuss exactly why the proposed system can protect the user (each voter)’s identity eveb with the user’s biometrics information.//
Author Response
Comment 1: Title: Machine learning is missleading: The revision may chance to Face recognition or Biometrics
Response:Many thanks for the comment. We have made modifications to the title of the paper using “A Publicly Verifiable E-voting System Based on Biometrics”.
Comment 2: The comparison Table-2 is not fair: Ref-[24] discusses Coercion resistance of e-voting system, whereas the authors’ submission does not. Revise Table-2 for being Fair. Does the authors’ proposal achieve coercion resistance?
Response:Many thanks for the comment. In response to your feedback, we have made modifications to Table 2 and added a new column to discuss the nature of coercion resistance.
Comment 3: Discussion on Related works with references is not enough. Add more related works on e-voting with biometrics including Face-recognition.
E.g
https://dl.acm.org/doi/10.1016/j.tcs.2022.05.005
https://www.sciencedirect.com/science/article/abs/pii/S0304397522002869
The authors shall review more related papers like above by authors themselves.
Response:Many thanks for the comment. We have added a section in the Related Works section to discuss the use of biometric technology for electronic voting, which will enrich the discussion of related works in the Related Works section.
Comment 4: Discuss exactly why the proposed system can protect the user (each voter)’s identity eveb with the user’s biometrics information.//
Response:Many thanks for the comment. We have added a new paragraph in the conclusion section to explain why this system can protect the identity of each user through their biometric information.
Reviewer 2 Report
Comments and Suggestions for Authors
The main concept of this article is a publicly verifiable E-voting system based on machine learning. The system is designed to address the limitations of traditional paper ballots and ensure transparency and trust in the voting process. The system uses convolutional neural networks for face recognition and zero-knowledge proofs for verifiability.
Figure 5 does not indicate the time unit.
Line 666 mentioned the figure 6, however, figure 6 is not in this article.
Please check carefully and correct all the typos and format in the revision.
Comments on the Quality of English LanguageEnglish could be further improved, and all the grammar and composition mistakes should be corrected in the revision.
Author Response
The main concept of this article is a publicly verifiable E-voting system based on machine learning. The system is designed to address the limitations of traditional paper ballots and ensure transparency and trust in the voting process. The system uses convolutional neural networks for face recognition and zero-knowledge proofs for verifiability.
Comment 1: Figure 5 does not indicate the time unit.
Response:Many thanks for the comment. In response to your feedback, we have made modifications to address the issue of not indicating time units in Figure 5.
Comment 2: Line 666 mentioned the figure 6, however, figure 6 is not in this article.
Response:Many thanks for the comment. This is indeed an error, and we have now changed Figure 6 at the error location by using the Figure 4.
Comment 3: Please check carefully and correct all the typos and format in the revision.
Comments on the Quality of English Language
English could be further improved, and all the grammar and composition mistakes should be corrected in the revision.
Response:Many thanks for the comment.We have made revisions to address typos and formatting in the text by making an undergone English language editing by MDPI.
Reviewer 3 Report
Comments and Suggestions for Authors
This article is a description of a publicly verifiable E-voting system based on machine learning. The system is designed to address the limitations of traditional paper ballots, such as temporal and spatial limitations and data security issues. The innovations of this system include the use of machine learning algorithms to ensure verifiability, the ability for voters to verify their own votes, and the use of a distributed ledger to ensure transparency and trust in the voting process. The content of the article is quite substantial, but the following issues need to be noted:
1. In the Introduction part of the article, the author introduces the writing background and raises many questions. However, there is no summary on how to solve these problems. This will cause a lot of difficulty for the reader to understand, because the reader needs to find the answer on his or her own.
2. 1.1 The Related works section only briefly lists the references and lacks a summary. Note that the role of Related works is to highlight the shortcomings of existing research, thereby demonstrating the innovation of the article's research.
3. It is recommended that the author directly lists the innovative points of the article. This improves the readability of the article.
4. It is recommended to separate the Related works part into an independent chapter.
5. Is it necessary for the 1.2 Organizations part to exist independently? I suggest the author consider it. I think it can be placed after the background introduction. So after this introduction, the research plan will be proposed immediately.
Author Response
This article is a description of a publicly verifiable E-voting system based on machine learning. The system is designed to address the limitations of traditional paper ballots, such as temporal and spatial limitations and data security issues. The innovations of this system include the use of machine learning algorithms to ensure verifiability, the ability for voters to verify their own votes, and the use of a distributed ledger to ensure transparency and trust in the voting process. The content of the article is quite substantial, but the following issues need to be noted:
Comment 1: In the Introduction part of the article, the author introduces the writing background and raises many questions. However, there is no summary on how to solve these problems. This will cause a lot of difficulty for the reader to understand, because the reader needs to find the answer on his or her own.
Response:Many thanks for the comment. We have summarized the solutions in the introduction section such that readers can read them more clearly.
Comment 2: 1.1 The Related works section only briefly lists the references and lacks a summary. Note that the role of Related works is to highlight the shortcomings of existing research, thereby demonstrating the innovation of the article's research.
Response:Many thanks for the comment. We have summarized the related Work section, pointed out the shortcomings of existing research, and proposed our innovative points for this paper.
Comment 3: It is recommended that the author directly lists the innovative points of the article. This improves the readability of the article.
Response:Many thanks for the comment.At the end of the second section of the paper, we directly proposed the innovative points to improve the readability of the paper.
Comment 4: It is recommended to separate the Related works part into an independent chapter.
Response:Many thanks for the comment. We have separated the related works section into a separate chapter in the revised manuscript.
Comment 5: Is it necessary for the 1.2 Organizations part to exist independently? I suggest the author consider it. I think it can be placed after the background introduction. So after this introduction, the research plan will be proposed immediately.
Response:Many thanks for the comment. We have placed the organizational part after the background introduction and cancel its independent existence.
Reviewer 4 Report
Comments and Suggestions for Authors
The paper A publicly verifiable E-voting system based on machine learning deals with an E-voting system design and its potential parameters. The presented system is based on commitment, machine learning and advanced algorithms in order to cover the most important aspects and features. The authors show that their system achieves privacy, counting hidden and verifiability and based on the evaluation, the system can be applied in practice.
First, the English level throughout the entire paper could be improved. I noticed several mistakes and typos in the text, therefore I recommend to perform some proofreading.
Next, although the topic is well covered, I find the overall article confusing and badly organized. First, it should be clearly described, what has been done, what are major benefits and pros of the designed system. Next, a comparison with existing solutions and the review of current approaches should be provided.
Next, the major part containing the description of the proposed system should be improved significantly. The description of deigned algorithms, protocols and processes should be presented using standard mathematical symbols and quotations. The current text description is mostly confusing, hard to follow and unnecessarily long.
Finally, a simple but extensive analysis and comparison of the designed system and existing techniques should be provided. The current text-based description is again hard to follow and pointlessly long.
Comments on the Quality of English LanguageThe English level throughout the entire paper could be improved. I noticed several mistakes and typos in the text, therefore I recommend to perform some proofreading.
Author Response
The paper A publicly verifiable E-voting system based on machine learning deals with an E-voting system design and its potential parameters. The presented system is based on commitment, machine learning and advanced algorithms in order to cover the most important aspects and features. The authors show that their system achieves privacy, counting hidden and verifiability and based on the evaluation, the system can be applied in practice.
Comment 1: First, the English level throughout the entire paper could be improved. I noticed several mistakes and typos in the text, therefore I recommend to perform some proofreading.
Response:Many thanks for the comment. We have made revisions by making an undergone English language editing by MDPI.
Comment 2: Next, although the topic is well covered, I find the overall article confusing and badly organized. First, it should be clearly described, what has been done, what are major benefits and pros of the designed system. Next, a comparison with existing solutions and the review of current approaches should be provided.
Response:Many thanks for the comment. We added the work content, advantages, and advantages of this system at the beginning of the paper, and compared it with other systems in the paper.
Comment 3: Next, the major part containing the description of the proposed system should be improved significantly. The description of deigned algorithms, protocols and processes should be presented using standard mathematical symbols and quotations. The current text description is mostly confusing, hard to follow and unnecessarily long.
Response:Many thanks for the comment.We have made modifications and improvements to the main part.
Comment 4: Finally, a simple but extensive analysis and comparison of the designed system and existing techniques should be provided. The current text-based description is again hard to follow and pointlessly long.
Response:Many thanks for the comment. We have added a simple and extensive analysis and comparison of the designed system and existing technologies in the sixth section of the paper.
Round 2
Reviewer 1 Report
Comments and Suggestions for Authors
Objection with comments to the revision
>Comment 2: The comparison Table-2 is not fair: Ref-[24] discusses Coercion resistance of e->voting system, whereas the authors’ submission does not. Revise Table-2 for being Fair. Does >the authors’ proposal achieve coercion resistance?
>Response Many thanks for the comment. In response to your feedback, we have made >modifications to Table 2 and added a new column to discuss the nature of coercion resistance.
The revision gives nothing on the definition nor explanation of the “coercer” issue of e-voting.
The revision does not cite the Ref [1] in the main boy of the paper.
The authors shall describe why the proposal fails to achieve the “receipt-free” or “coercion resistance”. Note that some existing works claim the receipt freeness in e-voting even with Biometircs, which are surveyed first by the authors themselves.
>Comment 3: Discussion on Related works with references is not enough. Add more related >works on e-voting with biometrics including Face-recognition.
>Response Many thanks for the comment. We have added a section in the Related Works >section to discuss the use of biometric technology for electronic voting, which will enrich the >discussion of related works in the Related Works section.
Indeed, the revision adds a section with related works, however, the revision is not clear what the challenging issues after the exiting works,
>Comment 4: Discuss exactly why the proposed system can protect the user (each voter)’s >identity even with the user’s biometrics information.//
>Response Many thanks for the comment. We have added a new paragraph in the conclusion >section to explain why this system can protect the identity of each user through their biometric >information.
This discussion should be given as a kind of Proof-style, while the added new paragraph in the conclusion is not enough nor rigorous.
Need more literature survey
[S-1]e-voting with biometrics including face recognition.
https://ijsrset.com/IJSRSET229314
https://easychair.org/publications/preprint/F65G
https://iopscience.iop.org/article/10.1088/1742-6596/1770/1/012011
[S-2]e-voting bulletin board
https://eprint.iacr.org/2021/1201
https://eprint.iacr.org/2020/109
https://eprint.iacr.org/2018/567
[S-3]Bulletin board based on Blockchain.
https://eprint.iacr.org/2021/047
https://iohk.io/en/research/library/papers/uncontrolled-randomness-in-blockchains-covert-bulletin-board-for-illicit-activity/
[S-4]e-voting based on biometrics and blockchain
https://ieeexplore.ieee.org/document/9425815
https://www.iajit.org/upload/files/An-On-Site-Electronic-Voting-System-Using-Blockchain-and-Biometrics.pdf
https://research.vit.ac.in/publication/an-authenticated-e-voting-system-using-biometrics-and-blockchain
https://typeset.io/papers/an-authenticated-e-voting-system-using-biometrics-and-1egkxvzvmt
[S-4] Receipt-fee and/or coercion resistance
https://eprint.iacr.org/2023/1509
https://research.birmingham.ac.uk/en/publications/beleniosrf-a-non-interactive-receipt-free-electronic-voting-schem
https://ietresearch.onlinelibrary.wiley.com/doi/full/10.1049/iet-ifs.2017.0213
The suggested references are just from the reviewer, but more related works should be checked by the authors themselves. Even some of MDPI-papers have survey on e-voting.//
Author Response
Comment 2: The comparison Table-2 is not fair: Ref-[24] discusses Coercion resistance of e->voting system, whereas the authors’ submission does not. Revise Table-2 for being Fair. Does >the authors’ proposal achieve coercion resistance?
>Response Many thanks for the comment. In response to your feedback, we have made >modifications to Table 2 and added a new column to discuss the nature of coercion resistance.
The revision gives nothing on the definition nor explanation of the “coercer” issue of e-voting.
The revision does not cite the Ref [1] in the main boy of the paper.
The authors shall describe why the proposal fails to achieve the “receipt-free” or “coercion resistance”. Note that some existing works claim the receipt freeness in e-voting even with Biometircs, which are surveyed first by the authors themselves.
Response:
Thanks for your opinion. We have proved that the work of this paper has also achieved Coercion resistance, mainly based on the following aspects:
(1) Privacy (Ballot Secrecy): Protect voters’voting information from being leaked to prevent anyone from tracking or identifying their votes. In this electronic voting system, voters' voting information cannot be tracked or checked.
(2) Verifiability: Provide a mechanism that allows voters to verify whether their votes were correctly recorded and calculated without revealing the content of their actual votes. In this electronic voting system, the verifiability of voting is achieved by using commitment and zero-knowledge proof methods.
(3) Non-coercibility: The system is designed to ensure that if forced, voters cannot provide the evidence required by the coercer to prove their voting method. In this electronic voting system, cryptography-based technology achieves this.
(4) Selective Disclosure: Allow voters to selectively disclose their voting information in order to prove their vote when verification is required, rather than being forced to disclose it when they do not want to. In this electronic voting system, due to the use of zero-knowledge proof and commitment technologies, voters can selectively disclose their voting information to prove their votes when verification is required.
Based on the above properties, this paper also achieves coercion resistance. We also explained and revised it in the paper, and strictly defined Coercion resistance (line 645). In addition, the reference [1] in the main text of the paper is cited in the introduction section (line 19).
Thanks again for your guidance.
>Comment 3: Discussion on Related works with references is not enough. Add more related >works on e-voting with biometrics including Face-recognition.
>Response Many thanks for the comment. We have added a section in the Related Works >section to discuss the use of biometric technology for electronic voting, which will enrich the >discussion of related works in the Related Works section.
Indeed, the revision adds a section with related works, however, the revision is not clear what the challenging issues after the exiting works.
Response:
Thanks for your opinion. At the end of the "Related Work" section (line 150), we summarize the existing work, point out the problems existing in the existing work, and propose the main content of this paper.
Thanks again for your guidance.
>Comment 4: Discuss exactly why the proposed system can protect the user (each voter)’s >identity even with the user’s biometrics information.//
>Response Many thanks for the comment. We have added a new paragraph in the conclusion >section to explain why this system can protect the identity of each user through their biometric >information.
This discussion should be given as a kind of Proof-style, while the added new paragraph in the conclusion is not enough nor rigorous.
Need more literature survey
[S-1]e-voting with biometrics including face recognition.
https://ijsrset.com/IJSRSET229314
https://easychair.org/publications/preprint/
Response:
Thanks for your opinion. We added a new section to the Security Proof section (line 667) to demonstrate why the system can protect each user’s identity through their biometric information.
Thanks again for your guidance.
Reviewer 3 Report
Comments and Suggestions for Authors
The main content of the article is the development of a publicly verifiable E-voting system based on biometrics, which aims to address the limitations of traditional paper ballots in ensuring voting rights and the transparency of election results. The system involves two phases: initialization and voting. In the initialization phase, each counter generates their own public and private key and publishes the public key on the bulletin board. In the voting phase, voters use their biometric information to cast their votes, which are encrypted and stored on the bulletin board. The system ensures verifiability through the use of a commitment scheme, zero-knowledge proof, and validation by a judge.
The author carefully revised the questions raised and highlighted the innovative points of the article, enabling the overall research content to be better presented. The innovation points of the article include the use of biometric information for voting, the publicly verifiable nature of the system, and the incorporation of cryptographic techniques to ensure the security and transparency of the election process.
Author Response
The main content of the article is the development of a publicly verifiable E-voting system based on biometrics, which aims to address the limitations of traditional paper ballots in ensuring voting rights and the transparency of election results. The system involves two phases: initialization and voting. In the initialization phase, each counter generates their own public and private key and publishes the public key on the bulletin board. In the voting phase, voters use their biometric information to cast their votes, which are encrypted and stored on the bulletin board. The system ensures verifiability through the use of a commitment scheme, zero-knowledge proof, and validation by a judge.
The author carefully revised the questions raised and highlighted the innovative points of the article, enabling the overall research content to be better presented. The innovation points of the article include the use of biometric information for voting, the publicly verifiable nature of the system, and the incorporation of cryptographic techniques to ensure the security and transparency of the election process.
Response:
Thank you very much for your comments.
Reviewer 4 Report
Comments and Suggestions for Authors
Some revisions were performed, however in my oppinion, the mathematical definition and description is still not very clear and consistent.
Author Response
Some revisions were performed, however in my oppinion, the mathematical definition and description is still not very clear and consistent.
Response:
Thanks for your opinion. There are indeed inconsistencies in mathematical definitions and descriptions in the paper. Please allow me to illustrate with examples. For example, there are two mathematical definitions of Pedersen commitment in the system description part, namelyand. The reason for this difference is because:
(1) is the initial commitment made by the voter;
(2) is the summary of commitments conducted by the vote counter;
Although the final conclusions drawn by the two may be consistent without being attacked, we adopt this expression in order to ensure rigor. I'm sorry that our description in the article is not clear enough. Thank you very much for your guidance and opinions. We have added corresponding explanations to the formula list (Table 1) in the article to make it clearer for readers.
Thanks again for your guidance.
